# The Long-Term Mechanical Properties of BS Perpendicular to the Grain

**DOI:** 10.3390/polym15010128

**Published:** 2022-12-28

**Authors:** Xiangya Luo, Xiangqian Luo, Haiqing Ren, Shuangbao Zhang, Yong Zhong

**Affiliations:** 1Research Institute of Wood Industry, Chinese Academy of Forestry, Beijing 100091, China; 2College of Materials Science and Technology, Beijing Forestry University, Beijing 100091, China; 3China Shanxi Sijian Group Co., Ltd., Taiyuan 030006, China

**Keywords:** bamboo scrimber, perpendicular to grain, duration of load effect, creep, DOL factor

## Abstract

As a modern bamboo composite with good mechanical properties, bamboo scrimber (BS) has achieved prominence in the sustainable architecture field. When used as a structural material, it is inevitably under continual tension perpendicular to the grain, therefore its mechanical response under long-term loading is significant for structural design. In this study, tensile tests were conducted on BS under short-term and long-term loads perpendicular to the grain. The duration of load (DOL) effect on BS perpendicular to grain and its creep effect were analyzed. Compared with BS parallel to the grain, the DOL effect on BS perpendicular to the grain was less severe, and the capacity for creep resistance was weaker. The threshold stress ratio and relative creep strain of BS perpendicular to the grain were 0.40 and 0.87, respectively. It was found that the DOL models and the viscoelastic model accurately predicted the DOL factor and creep strain. This study provides a scientific reference for the safe lifetime service of BS in practical engineering.

## 1. Introduction

Bamboo scrimber (BS) is a modern bamboo-based composite used in engineering, manufactured by mechanical treatment without separating the inner and outer layers of bamboo, achieving utilization above 90% [1]. As a renewable raw material, bamboo grows extremely quickly with a maximum growth rate of 1200 mm per day and is suitable for harvest every 3–5 years [2]. The graded vascular bundles that are distributed in full-pole bamboo are redistributed in bamboo scrimber [3]. In general, bamboo scrimber has become popular and shows further potential for use in green buildings due to its high and stable mechanical properties, good dimensional stability, and overall sustainability [4].

With the development of BS, it seems unavoidable in engineering practice that BS should be continually loaded in tension perpendicular to the grain. The mechanical response of BS subjected to continuous load includes a strength decrease known as the DOL effect and a deformation increase called creep. There have been no studies focused on the DOL effect or creep of BS perpendicular to the grain, although these are key indicators to ensure its safe lifetime service within structures. It remains unclear whether the DOL and creep effect of BS perpendicular to the grain are the same as parallel to the grain. Glulam made of Douglas fir experienced more severe loss of strength than that predicted by the Madison curve under stepwise load perpendicular to the grain [5]. Similarly, glulam made of Nordic spruce also had a more severe DOL effect perpendicular to the grain in different climate conditions than the Madison curve suggested [6]. However, the DOL effect was less severe than the Madison curve for the glulam made of SPF perpendicular to grain [7]. The creep factor for 50 years of SPF glulam was 2.158 perpendicular to the grain, which was greater than 0.576 for parallel to the grain [7].

This study therefore focused on the DOL and creep effect of BS perpendicular to the grain. Long-term and short-term tension tests were conducted. To obtain the DOL factors of BS perpendicular to the grain in terms of service years, the Foschi and Yao model and the Gerhards model based on accumulated damage theory and the Nielsen model based on fracture mechanics were developed. Furthermore, the Findley model and a viscoelastic model were employed to predict the creep factors.

## 2. Materials and Methods

### 2.1. Materials

The manufacturing process of BS contained five steps, as shown in Figure 1: splitting the full-pole bamboo, fluffing of bamboo tubes, steam heating, impregnation with resin, and hot pressing. Moso bamboo (*Phyllostachys edulis*) aged 3–5 years with an air-dried density of 0.74 g/cm^3^ was chosen for the manufacture of the BS. Firstly, the full-pole Moso bamboo was split parallel to the grain into tubes (Figure 1b) which were then fluffed parallel to the grain (Figure 1c). The fluffed bundles were steam-heated at 180 °C for 1.5 h then immersed in phenol formaldehyde resin with a weight ratio of 17% (PF 162510, Beijing Dynea Chemical Industry Co., Ltd., Beijing, China) for 15 min. After drying for 2 h, the bamboo bundles were laid in parallel and pressed at 137 °C for 20 min under a pressure of 4.0 MPa (Figure 1d). The formed BS panels (Figure 1e) were 2500 mm in length, 1300 mm wide, and 20 mm in height.

The BS specimens perpendicular to the grain were produced from the above panels according to the standard LY/T 3194-2020. Specimens from the same panels were assigned into four different test groups and labelled *A*, *B*, *C*, and *D*, with the first two groups used for the short-term tests and the latter two for the long-term tests. Each group contained 48 specimens made from different panels. All the specimens were moisture conditioned at 20 °C with relative humidity of 65%. At equilibrium, the moisture content of BS was 9.5% and the average density was 1.2 g/cm^3^. The dimensions of short-term and long-term test specimens are shown in Figure 2. According to the sample matching method, it was assumed that the cumulative probability distribution of short-term strengths for specimens in the long-term testing groups was identical to that in short-term testing groups, so the short-term strength of each long-term specimen was taken to be equal to that of the matched specimen of the same rank in the short-term testing groups.

### 2.2. Short-Term Test

The short-term tests for groups *A* and *B* were conducted using a mechanical testing machine (Model: 5582, Instron Co., Ltd., Norwood, MA, USA) referring to the standard LY/T 3194-2020. The loading rate was 1 mm/min which was equivalent to 403 MPa/h (Figure 3a).

The average strength was 141 MPa and the coefficient of variation (COV) was 14% for group *A*, while the comparative values were 135 MPa with a COV of 13% for group *B*. A least significant difference (LSD) test indicated that the average tensile strength between groups *A* and *B* had no significant difference (*p* = 0.124 > a = 0.05), and Levene’s tests for homogeneity of variance verified no significant difference in the variance of tensile strength between groups *A* and *B* (*p* = 0.467 > a = 0.05). Thus, the rationality of the above method of sample matching was demonstrated. Figure 3b presents a typical load–displacement curve from the short-term tests. It is apparent that the load increased linearly with the increase of displacement until the curve suddenly dropped, implying the specimen underwent brittle failure [8].

Figure 3c shows that the cumulative probability distributions of short-term strength for groups *A* and *B* were consistent, and the results of their combined fitting by the normal model, lognormal model, and 2-P Weibull model. The fitting results are listed in Table 1. The results indicate that the 2-P Weibull model fitted the data best due to its smaller values for both the maximum error of a single point and the sum of the squared error. The fitted size and shape parameters were 9.44 and 3.98, respectively. The estimated average strength was 8.61 MPa with a standard deviation (Std) of 2.41 MPa.

### 2.3. Long-Term Test

A lever system was designed and manufactured to conduct the long-term tests. The environment in the test lab was kept constant at 20 ± 2 °C with a relative humidity of 65 ± 5% (Figure 4a). The specimen was hung at one end of the lever and a load was applied using a certain weight of small steel cubes in a bucket that was hung at the other end of the lever with a leverage ratio (*l*_1_:*l*_2_) of 1:10, as shown in Figure 4b. The two testing groups *C* and *D* were tested simultaneously under a constant load of 5 MPa, i.e., the 8th percentile of short-term strength. Seven specimens in all were randomly selected for evaluation of the creep strain using dial gauges (Model: 2119S-10, Mitutoyo Co., Ltd., Tokyo, Japan), and each specimen was symmetrically equipped with two dial gauges of gauge length 85 mm, as shown in Figure 4c. Measurements of the creep strain were collected according to the protocol of once a minute for the first 5 min, then twice an hour, then after 48 h twice a day.

## 3. Results and Discussion

### 3.1. DOL Effect

#### 3.1.1. Test Results

The long-term test lasted for 5 months, during which 22 specimens failed in group *C* and 21 sin group *D*. Based on the assumptions of the sample-matching method, the long-term specimens were ranked according to failure time from short to long, so that the short-term strength of the *i*^th^ failed specimen could be estimated by the inverse function of the 2-P Weibull model: $\sigma_{s,i}=\mathrm{wblinv}\left(i/\left(n+1\right),9.44,3.98\right),\left(i=1,2,\ldots ,n\right)$. The stress ratio of the *i*th failed specimen was determined as: $S_{i}=\sigma_{c}/\sigma_{s,i}$.

The relationship curve between stress ratio and failure time reflects the DOL effect; the test data for groups *C* and *D* are shown in Figure 5. The DOL effect of BS perpendicular to the grain was compared with that of glulam perpendicular to the grain [9], glulam parallel to the grain [7], and the Madison curve [10]. The analysis revealed that the DOL effect of BS perpendicular to the grain was less severe than that of BS parallel to the grain. However, the DOL effect of glulam perpendicular to the grain was less severe than that of glulam parallel to the grain. These two opposite phenomena occurred mainly because the resin exerted a stronger positive influence on the tensile strength perpendicular to the grain compared with parallel. It was noted that the DOL effect of BS perpendicular to the grain was similar to that of the Madison curve and less severe than that of glulam parallel to the grain, again demonstrating the good long-term mechanical properties of BS perpendicular to the grain. Additionally, the convergent boundary for the DOL effect in BS perpendicular to the grain was estimated as 0.4.

#### 3.1.2. DOL Models

The complete picture of the DOL effect cannot be obtained from long-term tests alone, because test duration is always limited. Therefore, to predict the long-term strength various DOL models were developed based on different theories. the DOL models were applied to calculate the DOL factor of intended service years, which is beneficial for practical engineering. In this study, the Foschi and Yao model [11], the Gerhards model [12], and the Nielsen model [12] were applied to describe the DOL effect of BS perpendicular to the grain.

The Foschi and Yao model can be expressed as:

(1)$$\frac{d\alpha }{dt}=u_{1}{\left(\frac{\sigma \left(t\right)}{\sigma_{s}}-\eta \right)}^{u_{2}}+u_{3}{\left(\frac{\sigma \left(t\right)}{\sigma_{s}}-\eta \right)}^{u_{4}}\alpha \left(t\right),\;\;\;\sigma (t)>\eta \sigma_{s}$$
where *α* denotes the damage variable ranging from 0 to 1, *α* = 0 represents no damage in materials, *α* = 1 reflects failure; *u*_1_, *u*_2_, *u*_3_, and *u*_4_ are the model parameters that can be determined by regression of the test data; *σ*(*t*) is the long-term stress; *σ_s_* is the short-term strength; *η* is the threshold stress ratio below which materials are assumed not to fail under long-term stress. In this study, *η* was taken as 0.40 for BS perpendicular to the grain. Parameter *u*_1_ is dependent and can be expressed as:(2)$$u_{1}=\frac{k\left(u_{2}+1\right)}{\sigma_{s}{\left(1-\eta \right)}^{u_{2}+1}}$$
where *k* is the short-term loading rate, 403 MPa/h in this study. With *σ_c_* denoting the constant stress, S denoting the *σ_s_*/*σ_c_*, substituting Equation (2) into Equation (1), the time to failure (*t_f_*) can be obtained as follows:(3)$$t_{f}=\frac{\sigma_{c}}{k}+\frac{1}{u_{3}{\left(S-\eta \right)}^{u_{4}}}ln\left(\frac{1+\lambda }{\alpha_{0}+\lambda }\right)\;$$
(4)$$\lambda =\frac{k\left(u_{2}+1\right)}{u_{3}\sigma_{s}{\left(1-\eta \right)}^{u_{2}+1}}{\left(S-\eta \right)}^{u_{2}-u_{4}}$$
(5)$$\alpha_{0}={\left(\frac{S-\eta }{1-\eta }\right)}^{u_{2}+1}$$

2.The Gerhards model can be expressed as follows:

(6)$$\frac{d\alpha }{dt}=\exp(-\mu_{1}+\mu_{2}\frac{\sigma \left(t\right)}{\sigma_{s}})$$
where *µ*_1_ and *µ*_2_ are the parameters. The time to failure (*t_f_*) has the following expression, integrating Equation (6):(7)$$t_{f}=\exp(\mu_{1}-\mu_{2}\frac{\sigma_{c}}{\sigma_{s}})$$

3.The Nielsen model can be expressed as:

(8)$$\frac{dv}{dt}=\frac{{\left(\pi \cdot FL\right)}^{2}}{8q\gamma_{1}}\cdot \frac{v\cdot S^{2}}{{\left({\left(v\cdot S^{2}\right)}^{-1}-1\right)}^{1/\gamma_{2}}}$$(9)$$q={\left[0.5\left(\gamma_{2}+1\right)\left(\gamma_{2}+2\right)\right]}^{1/\gamma_{2}}$$
where *v* denotes the crack length divided by the initial crack length, *v* = 1 represents no damage, *v* = *S*^−2^ is full damage; *γ*_1_ and *γ*_2_ are the model parameters; *S* is the stress ratio; *FL* is the parameter reflecting the material quality. In this study, *FL* was taken as 0.2 for BS [13]. When *v* = *S*^−2^, *t_f_* can be obtained as follows:(10)$$t_{f}=\frac{8q\gamma_{1}}{{\left(\pi \cdot FL\cdot S\right)}^{2}}\int_{1}^{S^{-2}}\frac{{\left(\theta -1\right)}^{1/\gamma_{2}}}{\theta }d\theta$$
(11)$$\theta =\frac{1}{v\cdot S^{-2}}$$

The parameters of the above models were calibrated based on the experimental data of test groups *C* and *D*, as shown in Table 2. The coefficient of determination (*R*^2^) of the Gerhards model was 0.972, greater than 0.961 for the Foschi and Yao model and 0.931 for the Nielsen model. As shown in Figure 6, the fitting curves of the Gerhards model and the Foschi and Yao model were close prior to 50 days. In the first 50 days, the DOL effect of BS predicted by the Gerhards model and the Foschi and Yao model was less severe than that predicted by the Nielsen model. After 50 days, the DOL effect of BS predicted by the Nielsen model was between the predictions by the other two models. 

The predictions by the above models for the DOL factors, i.e., the stress ratios (*S*) of BS in different service durations are listed in Table 3, together with those of OSB [14], the Madison curve [10], glulam parallel to the grain [15], and glulam perpendicular to the grain [7]. The DOL factors of BS perpendicular to the grain predicted by the three models were all greater than 0.4, confirming the rationalization value of the threshold stress ratio (*ŋ*). The DOL factor of BS perpendicular to the grain was less than that of glulam perpendicular to the grain and the Madison curve but greater than that of BS parallel to the grain. The DOL factor in 50 years for BS perpendicular to the grain predicted by the Foschi and Yao model (0.549) is especially close to the value for glulam parallel to the grain (0.550).

### 3.2. Creep Behavior

#### 3.2.1. Creep Strain

The absolute creep strains of eight specimens are shown in Figure 7; it can be concluded that the creep strain experienced two stages of development. Initially, the creep strain increased at a diminishing speed for the first 4 days—we refer to this stage as the decelerated creep stage. Then, the creep strain developed stably and remained stable—this stage was the steady creep stage. The duration of the decelerated creep stage in BS perpendicular to the grain was much shorter than in BS parallel to the grain [16].

The absolute creep strain of BS perpendicular to the grain under constant load can be fitted using the Findley model as follows [17]:(12)$$\varepsilon (t)=m_{1}+m_{2}t^{m_{3}}$$
where *ε* is the absolute creep strain; *t* is the load duration (day); *m*_1_, *m*_2_, and *m*_3_ are the parameters that can be determined via regression analysis. The fitting results obtained by the model showed good predictive accuracy with determination coefficients of 0.95. 

Relative creep strain the generally used criterion to judge the creep resistance of materials. Based on the Findley model, the relative creep strain (*ε_r_*) can be calculated as follows:(13)$$\varepsilon_{r}=\left(\varepsilon -\varepsilon_{0}\right)/\varepsilon_{0}$$
where *ε*_0_ denotes the initial elastic strain. By substituting Equation (12) into Equation (13), the model can be expressed:(14)$$\varepsilon_{r}(t)=m_{4}t^{m_{3}}$$
where *m*_4_ = *m*_2_/*m*_1_.

The relative creep strain of eight BS specimens perpendicular to the grain together with the average fitting curve according to the Findley model are shown in Figure 8, as along with values for BS parallel to the grain [16], glulam perpendicular to the grain [9], and glulam parallel to the grain [7]. It was found that the relative creep strain of BS was smaller than that of glulam in the same grain direction, indicating that BS had stronger resistance to creep than glulam. The relative creep strain for 50-year loading duration, i.e., creep factor, in BS perpendicular to the grain was 0.87, about 40% of the value for glulam perpendicular to the grain. Moreover, the creep factor of BS in perpendicular was approximately 3.8 times that in parallel, similar to the relative values in glulam (3.7 times). The deformation-resistance ability in the tension of the interface between fibers and parenchyma determined the creep resistance of BS perpendicular to the grain, while the shear–slip interface ability decided that of BS parallel to the grain. The deformation-resistance ability under tension was weaker than the shear–slip resistance, causing the weaker creep resistance of BS perpendicular to grain.

#### 3.2.2. Viscoelastic Model

As a viscoelastic biomass material, the creep model for BS can be developed with different combinations of spring and dashpot, and the best model has been proven to be the modified Burgers model [18], as shown in Figure 9. 

The creep strain was calculated by Equation (15):(15)$$\varepsilon \left(t\right)=\varepsilon_{e}+\varepsilon_{ve}+\varepsilon_{v}=\beta_{1}+\beta_{2}(1-e^{-\beta_{3}t})+\beta_{4}t^{\beta_{5}}$$
where *ε_e_*, *ε_ve_*, *ε_v_* are respectively elastic, viscoelastic, and viscous strains; *β*_1_, *β*_2_, *β*_3_, *β*_4_, *β*_5_ are the model parameters.

The fitting results are shown in Figure 10. The determination coefficient (*R*^2^) for average creep strain was 0.99, indicating that the modified Burgers model provided a good prediction for the creep strain of BS perpendicular to the grain. 

Furthermore, the change laws of elastic, viscoelastic, and viscous strain in BS perpendicular to the grain were analyzed according to the modified Burgers model, as shown in Figure 11. The results demonstrated that the proportion of elastic strain gradually decreased in 50 years to 54.9%, because the elastic strain was constant. The proportion of viscoelastic strain increased in the first month and then continued to decrease, but the value remained above 9% and was much larger than 0~3% for BS parallel to the grain [16]. This indicates the increased viscoelasticity of the interface between fibers and parenchyma, or the interface between cells and resin, which play key roles in the creep of BS perpendicular to the grain. The proportion of viscous strain kept increasing with the passage of loading duration and increased to 34.1% in 50 years. Therefore, viscous strain made a significant contribution to the long-term creep strain of BS perpendicular to the grain, which was consistent with that of BS parallel to the grain.

## 4. Conclusions

The long-term mechanical properties of BS perpendicular to the grain were analyzed and compared with those of BS parallel to the grain. The main conclusions were drawn as follows:(1)The DOL effect of BS perpendicular the grain was less severe than in BS parallel to the grain and glulam parallel to the grain, but close to the Madison curve. The resistance to creep in BS perpendicular to the grain was weaker than in BS parallel to the grain. Consequentially, it should be considered which grain direction is appropriate to bear the long-term load in different construction circumstances.(2)The DOL factor in 50 years predicted by the Foschi and Yao model for BS perpendicular to the grain was 0.549, which is close to 0.550 for glulam parallel to the grain. Furthermore, the creep factor in 50 years for BS perpendicular to the grain was 0.87, about 40% that of glulam perpendicular to the grain. As a result, BS is considered a promising and appropriate substitute for wood composites in green buildings.(3)The Gerhards model presented a better fitting than the Foschi and Yao or Nielsen models. The modified Burgers model and the Findley model fitted the creep data well. Viscoelastic creep strain played a more significant role in BS perpendicular to the grain than parallel. Viscous strain contributed the most to the long-term creep strain in BS both perpendicular and parallel to the grain.

## Figures and Tables

**Figure 1 polymers-15-00128-f001:**
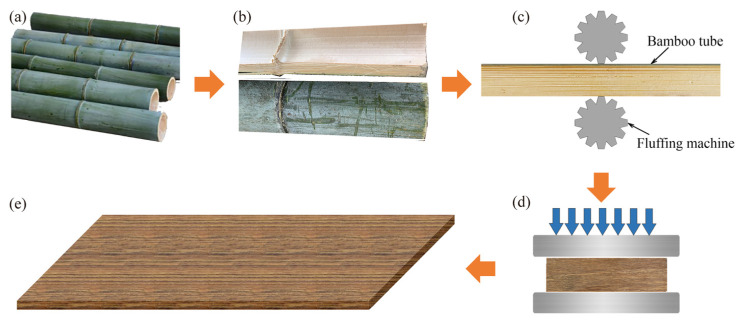
The manufacturing process of BS: (**a**) full-pole bamboo; (**b**) splitting; (**c**) fluffing; (**d**) hot pressing; (**e**) a bamboo scrimber panel.

**Figure 2 polymers-15-00128-f002:**
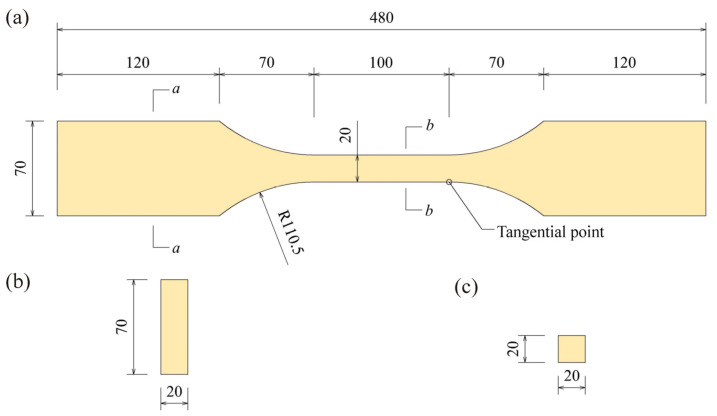
Dimensions of specimens perpendicular to grain in short-term and long-term tests: (**a**) whole specimen, (**b**) section a–a, (**c**) section b–b.

**Figure 3 polymers-15-00128-f003:**
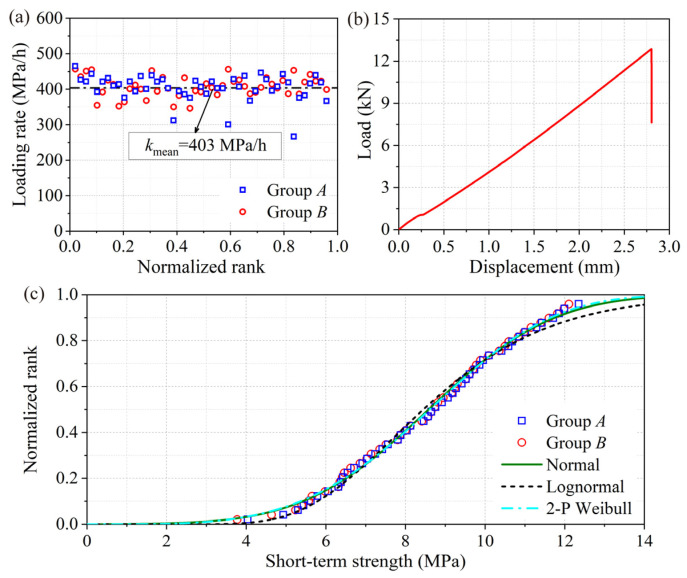
Results of short-term test: (**a**) the loading rate *k*, (**b**) a typical load–displacement curve, (**c**) the cumulative probability distribution of short-term strengths.

**Figure 4 polymers-15-00128-f004:**
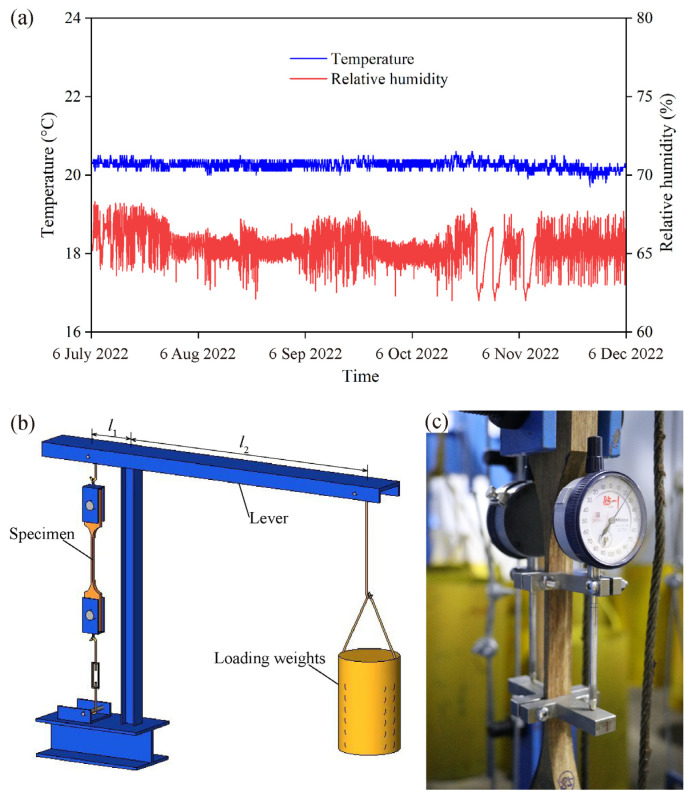
Long-term test: (**a**) constant environment; (**b**) schematic drawing of a lever system; (**c**) measuring creep strain.

**Figure 5 polymers-15-00128-f005:**
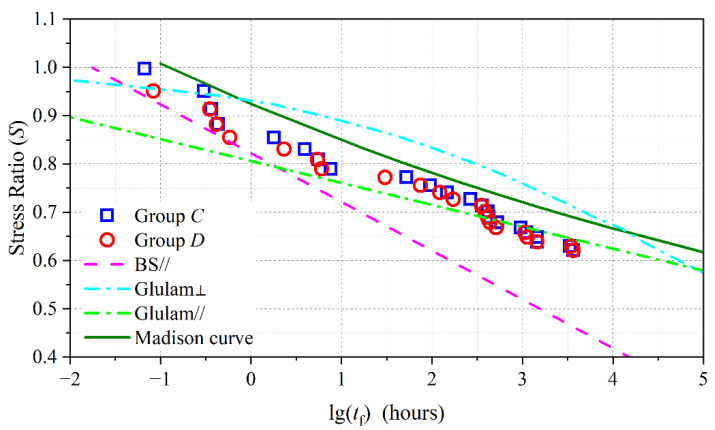
DOL effect of BS perpendicular to grain.

**Figure 6 polymers-15-00128-f006:**
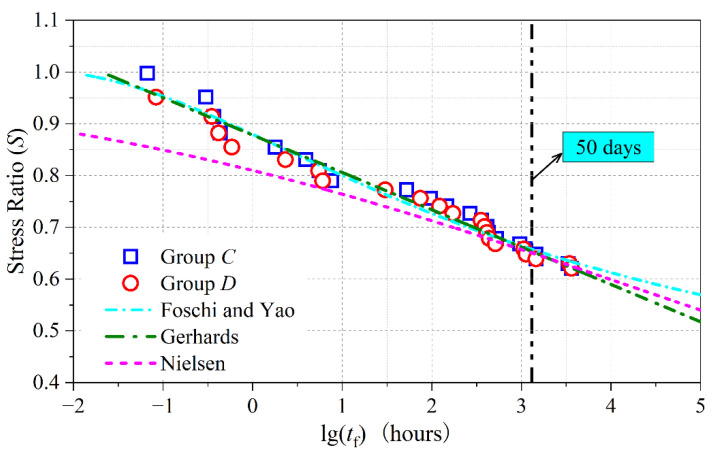
Fitted curves of DOL effect under constant load by different models.

**Figure 7 polymers-15-00128-f007:**
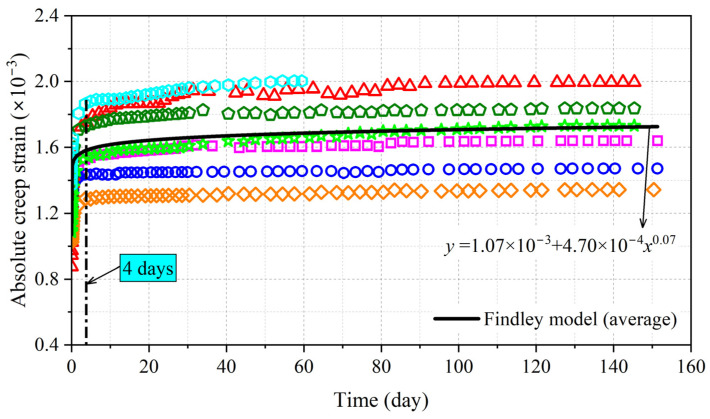
Absolute creep strain of BS perpendicular to grain.

**Figure 8 polymers-15-00128-f008:**
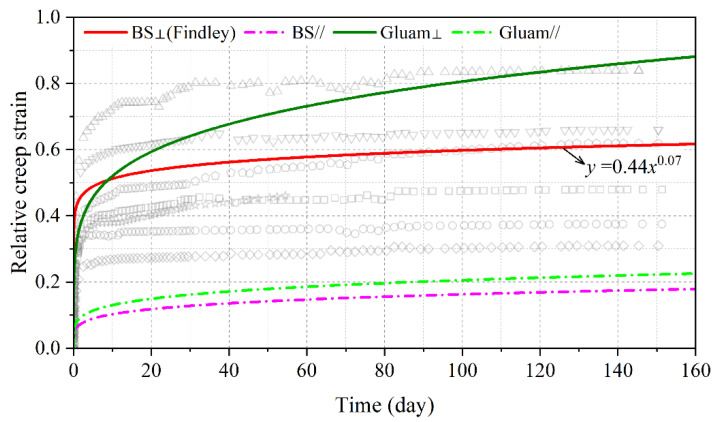
Relative creep strain of BS perpendicular to grain.

**Figure 9 polymers-15-00128-f009:**
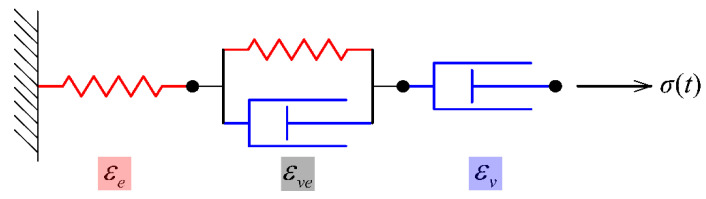
Modified Burgers model.

**Figure 10 polymers-15-00128-f010:**
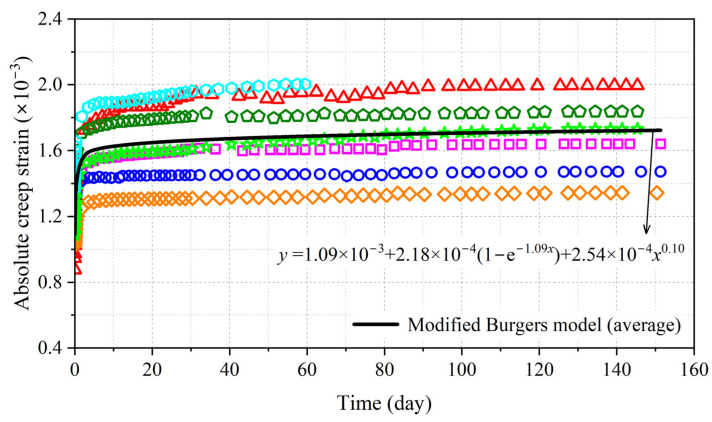
Fitting results by modified Burgers model.

**Figure 11 polymers-15-00128-f011:**
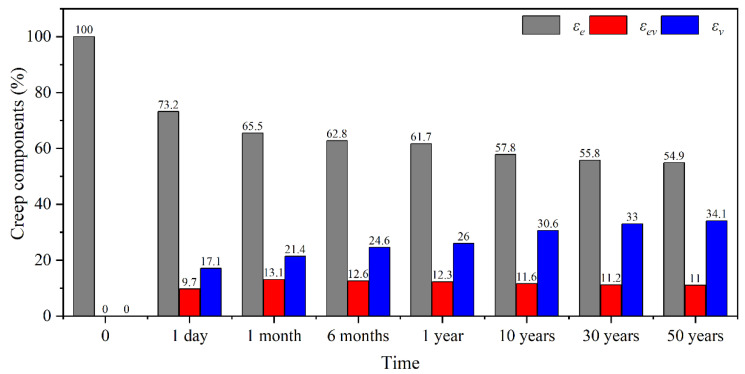
The change laws of creep components.

**Table 1 polymers-15-00128-t001:** Fitting results of short-term strength perpendicular to grain.

Probability Distribution Type	The Maximum Error of Signal Point	The Sum of Squared Error	Mean Value (MPa)	Std (MPa)
Test results	--	--	8.50	2.13
Normal	0.0416	0.0404	8.58	2.47
Lognormal	0.0761	0.1236	8.77	2.45
2-P Weibull	0.0366	0.0337	8.61	2.41

**Table 2 polymers-15-00128-t002:** Parameters of different DOL models.

DOL Model	Parameters	Mean	Std	R^2^
Foschi and Yao	*u* _2_	366.41	1.61 × 10^−3^	0.961
	*u* _3_	435.71	1.70	
	*u* _4_	3.72	1.99 × 10^−3^	
Gerhards	*µ* _1_	28.04	6.45 × 10^−1^	0.972
	*µ* _2_	31.92	8.43 × 10^−1^	
Nielsen	*γ* _1_	10.79	7.69 × 10^−3^	0.931
	*γ* _2_	0.154	6.71 × 10^−5^	

**Table 3 polymers-15-00128-t003:** DOL factors for different materials.

Service Years	BS⊥			BS//	Glulam⊥	Glulam//	Madison Curve
	Foschi and Yao	Gerhards	Nielsen				
5	0.584	0.538	0.525	0.397	0.629	0.595	0.635
10	0.572	0.515	0.501	0.385	0.613	0.582	0.620
30	0.554	0.480	0.464	0.367	0.592	0.560	0.599
50	0.549	0.463	0.447	0.360	0.583	0.550	0.589

## Data Availability

The data presented in this study are available on request from the corresponding author.

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
