# Peer review of "The Long-Term Mechanical Properties of BS Perpendicular to the Grain"

_polymers, 2022, doi:10.3390/polym15010128_

Round 1
Reviewer 1 Report
The authors would like to publish interesting article entitled "The long-term mechanical properties for BS perpendicular to grain". As structural material, bamboo scrimber (BS) answers loading under long-term tension, which has significant novelty. Moreover, the authors carried out tensile tests of BS perpendicular to grain under both short-term and long-term load were conducted. The duration of load (DOL) effect of BS perpendicular to grain as well as its creep effect is another significant feature of the article. Considered with those of BS parallel to grain, the DOL effect of BS perpendicular to grain was obtained by the author as less severe and the ability of creep resistance. Besides, it is very important that they accurately predicted the DOL models and viscoelastic model from the DOL factor and creep strain. For that reason, the journal can put the article into current archive.
Author Response
Following the reviewer's comment, we have had the English spell carefully examined. Meanwhile, we also re-examined the grammatical issues of the manuscript with great care. And thanks very much for the positive comments. More details please see the attachment.

Reviewer 2 Report
I find the topic very interesting and it also has to do with the tendency of getting "greener" advanced products which is very important. I would like your Conclusions part to be with more details and not reffering only to each method of analysing and its results but conclusions in general for these type of composite materials as a product in real life conditions using the results of the previous mathematical-statistical analysis.
Author Response
Thanks for the comments to help improve the quality of this paper. Following the suggestions, the English languages have been carefully checked and the introduction has been modified. Besides, more realistic and practical conclusions were added. More details please see the attachment. Thanks again for the comments and suggestions.
